# Polymeric Orthosis with Electromagnetic Stimulator Controlled by Mobile Application for Bone Fracture Healing: Evaluation of Design Concepts for Medical Use

**DOI:** 10.3390/ma15228141

**Published:** 2022-11-17

**Authors:** Filipe Bueno Vilela, Eduardo Serafim Silva, Mirian de Lourdes Noronha Motta Melo, Rochelly Mariana Pedroso Oliveira, Patricia Capellato, Daniela Sachs

**Affiliations:** 1Centre for Studies and Innovation in Biofunctional Advanced Materials, Institute of Physics and Chemistry, Unifei-Federal University of Itajubá, Av. BPS, 1303, Itajubá 37500-903, MG, Brazil; 2Institute of Mechanical Engineering, Unifei-Federal University of Itajubá, Av. BPS, 1303, Itajubá 37500-903, MG, Brazil

**Keywords:** tissue engineering, rehabilitation engineering, polymeric orthosis, additive manufacturing, electromagnetic stimulation, bone healing

## Abstract

**Background:** The occurrence of bone fractures is increasing worldwide, mainly due to the health problems that follow the aging population. The use of additive manufacturing and electrical stimulators can be applied for bioactive achievements in bone healing. However, such technologies are difficult to be transferred to medical practice. This work aims to develop an orthosis with a combined magnetic field (CFM) electrostimulator that demonstrates concepts and design aspects that facilitate its use in a real scenario. **Methods:** A 3D-printed orthosis made of two meshes was manufactured using PLA for outer mechanical stabilization mesh and TPU for inner fixation mesh to avoid mobilization. A CFM stimulator of reduced dimension controlled by a mobile application was coupled onto the orthosis. The design concepts were evaluated by health professionals and their resistance to chemical agents commonly used in daily activities were tested. Their thermal, chemical and electrical properties were also characterized. **Results:** No degradation was observed after exposure to chemical agents. The CMF achieved proper intensity (20–40 µT). The thermal analysis indicated its appropriate use for being modelled during clinical assessment. **Conclusion**: An orthosis with a coupled electrostimulator that works with a combined magnetic field and is controlled by mobile application was developed, and it has advantageous characteristics when compared to traditional techniques for application in real medical environments.

## 1. Introduction

The occurrence of bone fractures is increasing worldwide, mainly due to the health problems that follow the aging population, with average life expectancy becoming progressively higher. This fact points out the significant socioeconomic impact on global health systems. According to data from the United Nations research, the population over 65 years old will elevate from 524 million in 2010 to approximately 1.5 billion in 2050 [1,2]. Therefore, with more people getting older, the probability of fractures occurring, especially in leg, wrist or hip bones, also increases [3,4,5]. Statistically, the prevalence of bone fracture occurrences is related to the upper limb region, specifically in the distal area of the radius and metacarpal, accounting for an incidence of 29.2% of the total cases. Subsequently, there are a lower number of limb fractures in the ankle and metatarsal region (14.8%) and femur (11.6%) [6].

After the fracture, the use of immobilization apparatus is crucial to align the bone’s extremities. Orthopedic plaster (CaSO_4_.2H_2_O) and fiberglass cast are the most common materials used in orthopedic immobilization, mainly because of their ease of acquisition and low cost. However, these materials are correlated to clinical complications, such as microbial activity on the dermis, with occurrence of skin irritation, deep vein thrombosis, and compartment syndrome due to the increased pressure in the region and difficulty in defining fixation points for fracture stabilization [7,8]. Thus, some advanced materials have been proposed in order to avoid these problems and ensure the customization of orthoses. These advanced materials have specific properties, such as responsiveness to external stimuli (e.g., temperature) and shape memory, in order to achieve customizable solutions for each patient and clinical condition. The application of additive manufacturing methods and tridimensional (3D) printers has been used for providing a more accurate stabilization of fractures, reducing the risks mentioned before [9]. Comparisons made in the literature with 3D orthotics, mainly made of polylactic acid (PLA), thermoplastic polyurethane (TPU) and acrylonitrile butadiene styrene (ABS), and traditional immobilization techniques show that the use of additive manufacturing allows for better customization, as well as higher patient acceptance and engagement in treatment [1,10,11,12]. Nonetheless, just like orthopedic plaster and fiberglass cast, most 3D-printed orthoses are biologically inert. In other words, they are not able to positively optimize the patient’s welfare during the treatment and speed up healing time. In addition, 3D-printed orthoses are reviewed in the literature as less appropriate than traditional methods for aspects such as: (a) difficulty in fully immobilizing, having areas of limb mobility within the orthosis when it is in a rigid state, (b) inaccuracy in establishing fixation points for bone alignment at the time of modeling, and (c) insecurity of health professionals in the use of novel materials and software for modeling [7,12].

Considering the inert behavior of the orthosis, the use of ultrasound or electrical stimulators can be applied for bioactive achievements. It is widely stated in the scientific literature that mechanical vibrations and electromagnetic fields used within the fracture area stimulate osteogenesis due to thermal and non-thermal effects. The thermal effect occurs by the friction of the vibrating molecules that raise the local temperature and increase the blood flow and flexibility of the collagen structures present [1]. As for the non-thermal effect, the applied energy generated makes the ions pass through the cytoplasmic membrane of the cells more easily, accelerating cellular metabolism [1,13]. The main mechanisms activated by the application of mechanical vibrations or electromagnetic fields for bone regeneration are: proliferation and differentiation of mesenchymal stem cells, which are cells capable of differentiating into other functional cells, such as osteoblasts; and stimulation of the regulation of bone morphogenetic proteins via the production of calcium-calmodulin (bone growth factors), phospholipase A2 (enzymes for breaking down fatty acids), synthesis of prostaglandin E2 (regulation of muscle tone) and other components that contribute to healing and bone callus formation [14,15].

The frequency and intensity parameters of stimulation must be controlled to achieve the benefits. Ultrasound devices commonly works with a frequency of 1–5 MHz and intensity of 2–3000 mW/cm^2^ for an application of 5–60 min/day [13]. For electromagnetic non-invasive devices, their different construction designs should be analyzed. One of the most efficient versions is the electromagnetic stimulator by combined magnetic field (CMF), because it shows healing effects by applications of 30 min/day within around 76.6 Hz of frequency and magnetic field intensity of 20–40 µT.15. Comparing the stimulation techniques and design features of devices, ultrasound ones show lower relevance (38–50%) to reduce the healing time in post-bone-fracture treatments than electromagnetic devices (70–98%). Moreover, CMF electromagnetic stimulators have drawn attention because they can achieve therapeutical effects with use for 30 min/day, while other electromagnetic techniques should be applied for 1–24 h/day to achieve the same results [15,16].

Despite the fact that the electromagnetic stimulation is beneficial, so far, a solution has not been found in the scientific literature that can concomitantly be integrated into a non-invasive electrostimulator device of small dimensions or applied to a printed orthosis used in immobilization and bone fracture rehabilitation. The possibility of integrating these two systems of immobilization and stimulation is fundamental for their technological transfer. However, although several studies have dedicated attention to this area, few are focused on engendering a sufficient design that allows the use of technology in medical practice. Most studies analyze the proliferation effect of orthogenic cells from in vitro assays with laboratory generators of electric and magnetic fields, which is improper for patient use [17,18]. The studies that were dedicated to the development of electrostimulator devices are summarily focused on implantable devices. Such devices, however, provide risks inherent to the surgical process, such as inflammatory processes from material rejection and microbiological infection [19,20]. Furthermore, studies that use non-invasive electrostimulation devices acquire commercial systems, which were not designed for integration with orthopedic immobilization meshes. In this case, they have inadequate dimensions for everyday use, ranging from 28 to 31 cm in diameter, with a mass between 400 g and 900 g. In addition, in most cases, required applications must vary from 2 to 24 h/day, and they are stimulated with the application of electric current from electrodes positioned on the skin [21,22,23,24,25,26,27,28]. Electric shock hazards are intrinsic to this type of application. Thus, although the benefit of electrostimulation for bone regeneration is a consensus, the results are limited for the proposed systems to actually be used in practical ways, raising concerns about their use for the clinician and rejection by the patient [15,16,29,30].

Additionally, 3D prosthesis printing has not yet reached its practical relevance in medicine despite its benefits, among other factors, due to the low targeting of scientific solutions to practical problems, such as guaranteeing the stability of the bone fixation point even after deswelling of the hematoma and allowing the concomitant use with stimulation therapy. The progression of the clinical scenario makes it possible to observe that the printed orthoses still allow degrees of freedom of the patient’s articulation. Such movement is undesirable for ensuring fracture alignment during bone callus formation. Additionally, literature reviews indicate that new solutions must be proposed so that they can be used in a real environment for long-term therapy, allowing a comparison in real applications regarding their benefits against traditional techniques, such as orthopedic plaster [31,32,33].

Based on the above, this work presents the development of a biocompatible two-meshes polymeric orthosis for coating and ensuring immobilization of fractured areas of the radiocarpal joint with a coupled non-invasive device for electrostimulation by CMF. The electrostimulation and progress of treatment is controlled and displayed by a mobile application with system data stored in a remote database for clinical follow-up and patient engagement. The system developed and presented in this research work aims to show the use of accessible additive manufacturing techniques to produce an orthosis with properties seen as beneficial for clinical treatments, as well as characteristics that make its practical use in medicine possible. Therefore, the following definitions were incorporated: orthosis with double meshes, flexible and rigid, for filling and stabilizing the fractured region, avoiding any possibility of movement; electrostimulation transducer of reduced dimensions and with the possibility of integration within the orthosis; use of CMF for short-term applications; use of system control by mobile application and database to facilitate the monitoring of treatment by the doctor at a distance. Therefore, the study focuses on discoveries and materials consolidated in the scientific environment as suitable for biomechanical rehabilitation and proposes new formats, characteristics and technological integrations to enhance the possibility of medical clinic using the solution.

## 2. Materials and Methods

### 2.1. Orthosis Development

Polymeric samples of PLA, TPU and ABS were printed using a FDM printer (Cliever CL2 Pro+) and filament of 1.75 mm, at printing temperatures of 210, 220 and 240 °C, respectively. Moreover, a printing speed of 80 mm/sec, layer thickness of 0.8 mm, layer height of 0.1 mm and printing density of 75% were used.

The thermal analysis of all polymeric samples was investigated using a differential scanning calorimeter (DSC 60, Shimadzu). Samples (7.5–9.9 mg) were sealed in aluminum pans and heated from a room temperature of 25 °C to 250 °C at a rate of 5 °C/min under an inert (N2) atmosphere. The results obtained were treated using OriginPro 2021 software (OriginLab Corporation^®^, Northampton, MA, USA).

The hydrophilicity of the polymeric samples was evaluated through their contact angles. The method used was the sessile droplet, with the deposition of a 10 µL deionized droplet on the sanded surface of the material at room temperature (24 °C ± 1 °C). The equipment used for the analysis was the Kruss Easy Drop goniometer.

As a resource to be used daily and exposed to chemical agents, the orthosis was exposed to different chemical agents, particularly as it involves the wrist and hand region. Thus, an analysis was carried out to verify the occurrence of failure or some level of degradation of the polymers when exposed to water, sweat, alcohol and soap. Twenty samples were produced and immersed into 1 mL of each agent for a sampling of 5 units of each polymer. For the control group, 5 samples of each polymer were kept in empty wells. The chemical agents were obtained as follows: (i) sweat: artificially produced, using 500 mL of distilled water, 0.05 g of urea and 0.25 g of NaCl [34]; (ii) soap: commercial liquid detergent composed of linear sodium alkyl benzene sulfonate, triethanolamine alkyl benzene sulfonate, sodium lauryl ester sulfate, coco amido propyl betaine, magnesium sulfate, formaldehyde and water; (iii) alcohol: commercial alcohol gel 70° composed of hydrated ethyl alcohol, propylene glycol and aqueous vehicle; (iv) water: distilled water. The samples were kept in the wells for 40 days. At the beginning of the 41st day, the samples were removed from the wells, placed on a dry surface and relocated in empty wells for microscopic evaluation.

Then, the samples were investigated using Scanning Electron Microscope (SEM), having been sputtered with gold using Sputer Ion Coater (IC-50 Shimadzu) under a current of 8 mA in vacuum and 15 kV of acceleration voltage. Scanning was performed using Shimadzu SS550 SEM.

The characteristics of the orthosis and its comparison with traditional methods of immobilization and stimulation were shared with health professionals for their evaluation. The data were compiled into a document and submitted to the standard of the Research Ethics Committee (REC). The sampling of evaluations was *n* = 7, restricted to health professionals related to traumatology, including orthopedic doctors and physiotherapists specialized in the traumatology area. The possible answers for each concept of the orthosis and their numerical counterparts were: “completely agree” (3 points), “partially agree” (2 points), “partially disagree” (1 point), “completely disagree” (0 points) and “not able to evaluate” (no score considered for the calculation). Aspects with average grade equal or superior to 2 indicated features that should be kept in the final design.

The use of research as a way of defining the format and characteristics of the orthosis is a step considered fundamental for the scientific result to be closer to medical practice. The documentation was sent to the REC of the Universidade do Vale do Sapucaí (UNIVÁS) in Pouso Alegre, Brazil. It is detailed in the supplementary data annexed to this paper (Appendix A). The Certificate of Ethical Assessment Presentation in Plataforma Brasil is 40573320.9.0000.5102. Based on the interviews, the following technical aspects were raised for the production of the orthosis: use of a flexible mesh for patient comfort and guarantee of immobilization of the radiocarpal joint; use of rigid mesh overlapping the internal mesh for mechanical stability of the fracture; use of holes with rounded corners in the orthosis for local ventilation and to prevent dirt accumulation; format that can be easily adjusted in a computer program for different anthropometries; possibility of positioning an electrostimulation transducer, with the transducer position easily adjusted before printing for each clinical case. After formatting the meshes of the orthosis, the files were exported in the computational extension, STL. The Cura 4.8.0 software (Ultimaker BV) was used to slice the 3D object, so that it could be sent to the Cliever CL2 Pro+ printer. The orthosis was manufactured with dimensions of 40 × 58 mm for the inner fixation mesh and 190 × 180 mm for the outer stabilization mesh, with the possibility of adjusting the dimensions for anthropometric adaptation. The design bases presented in the literature for orthopedic orthoses were considered, which were then adapted and modified based on the evaluation of the health professionals [9,10,11,12].

### 2.2. CMF Stimulator Development

The microcontroller circuit used the ATMEGA16U2 model (Atmel), generating square pulse by Pulse Width Modulation (PWM) at 76.6 Hz. The signal was filtered to sinusoidal waves using a resistor-capacitor low-pass filter of second order and amplified by a current-gain transistor amplifier composed by a TIP122. The stimulation coil was produced by winding a nickel–copper alloy wire with a diameter of 0.16 mm, using 500 turns of the wire around an ABS apparatus. As the apparatus does not have direct contact with the patient’s skin, ABS was used because of its glass transition (~105 °C) temperature being higher than that of PLA (~60 °C) [35,36]. Thus, as part of the generated magnetic field energy is converted into heat, which could reach high levels during high-intensity applications, the use of ABS apparatus was used to avoid deformation.

To characterize the generated electrical signal, an oscilloscope model DSO1052B, 50 MHz, 1 Gsa/s, from Agilent Technologies^®^ was used. Furthermore, a magnetic field meter (teslameter), model PT2026 NMR (Metrolab^®^, Plan-les-Ouates, Switzerland), was used. For analyzing the CMF behavior, the alternated and constant fields were evaluated. The coil was positioned perpendicular to the support surface and aligned with the meter probe. The distance from the coil to the meter probe was analyzed when it measured the field intensity of 40 µT and 20 µT. Then, a second analysis was performed for the alternated magnetic field generation by the electrostimulator in an open field. The equipment used for this evaluation was the EHP50-TS field spectrum analyzer (Narda Safety Test Solutions^©^, Cisano sul Neva, Savona, Italy). The coil was positioned perpendicular to the EHP50-TS sensor and was energized with an alternating 76.6 Hz signal combined with the continuous signal (offset voltage).

For characterizing the possible influence of biological tissue on the propagation of the magnetic field up to the fracture, bovine biological tissue was acquired from the hip region, with a thickness of 3.1 cm in the thickest region and 2.2 cm thick in the thinner region, and it was 15.8 cm long. Bovine tissue was used in accordance with previous research works on biomechanics [37,38,39,40]. The coil was placed perpendicular to the support surface and aligned with the probe of the magnetic field meter for 5 records. The distances between the coil and the probe of the magnetic field meter were evaluated when it detected 20 µT and 40 µT, these being the reference values for the CMF effectiveness [15,16].

A system was developed to manage the number of applications performed, frequency and alerts for the correct time to apply electrostimulation. Therefore, a Bluetooth module (RS232 HC-05, 2,4 GHz) was included in the circuit, which sends and receives data from a smartphone application. The mobile application was developed using blocky Java-based language, on the Kodular^®^ platform. The collected data were stored in a remote Firebase^®^ database.

## 3. Results

Through the DSC thermal analysis, the glass transition temperature (Tg) and the melting temperature (Tm) of each polymer were obtained. PLA (61.97 °C/145.20 °C), ABS (102.52 °C/140.92 °C) and TPU (25.6 °C/168.2 °C) showed Tg behavior close to that specified in literature and commercial data [41,42,43]. For wettability results, it was observed that, with the exception of ABS, the polymers evaluated showed hydrophilic behavior (θ < 90°). 

The SEM analysis aimed to identify if the polymers were degraded under exposure to chemical agents. No evidence or characteristic forms of microbial proliferation were seen in any group. The images were compared with previous SEM works for printed polymers. Except for the common flaws observed in Figure 1, there were no unusual surface changes in any samples in all evaluated groups [44,45].

The health professionals analyzed the concepts of the system. The results of the survey carried out with them were registered within the REC, and the individual score of each question is presented in the Appendix A. Thus, the aspects considered relevant by the interviewees, totaling 88.89% of those presented in the questionnaire (aspects with grades equal or higher than 2), corroborated the characteristics defined for the orthosis, such as: (a) holes for ventilation of the covered area in order to reduce local humidity and accumulation of dirt and microorganisms; (b) composition of two overlapping meshes to ensure immobilization of the fractured area and facilitate the definition of fixation points; (c) use of temperature as a stimulus for the morphological alteration of the meshes for customization; (d) use of a non-invasive electromagnetic field and no direct current application to avoid risks; (e) integration of the electrostimulator into the orthosis as a differential for reducing the treatment time; (f) application of the CMF technique for 30 min daily of exposure, and (g) inclusion of a control system with mobile application to ensure engagement. The printed orthoses just after the manufacturing process and also after modeling over the wrist, with PLA outer mesh heated around 55–60 °C to become temporarily flexible, are shown in Figure 2.

For the electrostimulator, the response of the signal amplified by the TIP122 transistor circuit, configured as a common emitter, was measured from the collector, where the transducer coil was connected. The offset voltage is a result of the positive polarization of the generated signal to create the continuous magnetic field (0 Hz) concomitantly with the alternating magnetic field (76.6 Hz), in order to establish the CMF. Moreover, the system can have its frequency adjusted in order to allow comparisons for different frequency applications in bone healing for future studies in practical applications. 

The magnetic field generated because of the current in the conducting coil was analyzed regarding the field strengths (40 µT and 20 µT) in relation to distance, as shown in Table 1.

The test configuration for magnetic field permeability over biological tissue can be seen in Figure 3, where the biological tissue was interposed with the electrostimulator coil. No statistically significant changes were identified in the readings taken.

The analysis of magnetic field strength in open space was performed in the laboratory, connecting the EHP-50 magnetic field analyzer (Narda Safety Test Solutions^©^, Cisano sul Neva, Savona, Italy) via optical fiber to the computer, to reduce electromagnetic interference from the electrical signal conduction in the system. The equipment was adjusted for a spectrum of 0–100 Hz. The system detected, above the spurious field threshold (<0.1 µT), two prominent peaks. The highest field-intensity peak was identified at the frequency of 79.59 Hz with 41.14 µT of magnetic field strength. The second, lower-intensity peak was identified at a frequency close to 60 Hz with an intensity of 0.20 µT (Figure 4a). The first peak is the result of the magnetic field generated by the electrostimulator, while the second peak is a result of interference from the residual magnetic field generated by the electrical grid in the environment. To prove this interpretation, a second reading was taken with the electrostimulator turned off, away from the test area. The cancellation of the peak at 79.59 Hz and the presence of the peak at 60 Hz was observed, although at a lower intensity (<0.1 µT), as can be seen in Figure 4b.

After the development of the electrostimulator, its coil was integrated into the protrusion designed in the immobilization mesh of the orthosis. Integration took place without difficulties and the coil remained stuck even after moderate movement, since it was stuck to the slot present in the mesh. It was observed that the coil was positioned over the thinnest region of the fixation mesh, promoting a maximum distance between the coil and human skin of 3 mm. The electrostimulator circuit was protected in a commercially available polyvinyl chloride box. Figure 5a is an image taken from the coupling of the electrostimulator to the orthosis.

The mobile application used integrates with the electrostimulator coupled to the orthosis through wireless Bluetooth communication. Through the application, the physician can follow the patient’s history of stimulation to ensure that he/she is following the treatment as indicated. The physician can also set the application frequency to validate a different value from the standard (i.e., 76.6 Hz) for specific bone fracture investigation. Therefore, the mobile application has the following features, sequentially from the top to the bottom of the screen (Figure 5b): (a) display of the number of electrostimulation applications performed on the patient; (b) display of the frequency set by the clinician with frequency increase and decrease buttons, where the values vary by ±5 Hz when clicking on the buttons, with the exception of the increase from 70 Hz or decrease from 80 Hz, when the value will be set to 76.6 Hz; (c) frequency application button, which will set the frequency used by the electrostimulator per se; (d) display of Bluetooth list to connect the application with the electrostimulator; (e) display of resources for selecting the hour/minute of alert for the application of electrostimulation; (f) and alert activation or deactivation button. Figure 5b shows the mobile application and database registration of a patient.

## 4. Discussion

From the results obtained, it is possible to verify the adequacy of the solution within a design that includes innovative and potential aspects to facilitate the transfer to medical practice. The main aspects of innovation foreseen in the objective were incorporated as following: orthosis with double meshes, flexible and rigid, for filling and stabilizing the fractured region, avoiding any possibility of movement; electrostimulation transducer of reduced dimensions and with the possibility of integration within the orthosis; use of CMF for short-term applications; use of system control by mobile application and database to facilitate the monitoring of treatment by the doctor at a distance.

Regarding the choice of the polymers, it was made based on commercially biocompatible thermoplastic biomaterials that can be easily obtained. The additive manufacturing technique by extrusion is also considered a low cost and easy operation option among those currently available, which is another element that eases the incorporation of the technology into medical use [46]. Regarding the DSC analysis, TPU had the lowest Tg, being flexible under room temperature. Its flexibility was considered in the final design for an inner mesh over the skin, in order to avoid wrist movement in the orthosis, and it was used to facilitate the definition of bone fracture fixation areas, as critical points of 3D-printed immobilization solutions identified in previous studies [7,12]. PLA was considered as an outer mesh over the TPU one for mechanical stabilization. PLA has the lowest Tg between the polymers after TPU and presents proper mechanical properties, with tensile strength around 50–70 MPa and elastic modulus of 3.4 GPa [41,42]. The combination of the internal flexible mesh filling the space between the patient’s skin and the external rigid mesh aims to address a response to one of the most notorious problems identified in the literature for printed orthoses, which is the possibility of articulating the fractured bone [7,12]. In this way, such an arrangement of meshes can mitigate the clinical risks related to the non-alignment of the fractured bone.

ABS was just used for the transducer coil apparatus and not for the orthosis’ mesh because it is characterized as a typically hydrophobic material, confirmed by the contact angle test [47]. For non-invasive systems, the non-occurrence of allergic effect or skin irritation is a relevant aspect. In this perspective, scientific review studies indicate that the use of hydrophilic surfaces and solvents reduce the irritating potential of the material when in contact with the skin [48].

The analysis of chemical agents showed that the chosen biopolymers have suitable behaviors. None of the samples were degraded in the presence of chemical components such as alcohol, sweat and soap, even after exposure for more than 40 days. This result is relevant for the use of the orthosis in long-term applications, where such chemical elements are present during the patient’s daily activities. The presence of soap, water, sweat and alcohol are part of the group of main activities that involve upper limbs [49]. Thus, as no detachments of polymeric layers, grooves, deformations, release of particles or color change in the samples were identified, the orthosis is suitable for exposure to these chemical agents. Despite the importance of this assessment, no studies were identified in the scientific literature aimed at investigating the impact of exposure of printed biopolymers to chemical agents related to the daily life of the user [50,51,52].

Moreover, the concept of relative magnetic permeability for biological tissues and imperceptible change in signal intensity was proved. The result is consistent with that seen in the scientific literature, which indicates that biological tissues in general have relative magnetic permeability equal to 1, that is, the equivalent of permeability in a vacuum [53].

The electromagnetic response of the electroestimulator was studied to identify if it achieves the frequency and intensity recommended in the scientific literature. It is specified in the literature that the practicable values in the use of CMF for bone fracture regeneration are 40 µT in its alternating portion and 20 µT in its continuous one [16,54]. According to the data in Table 1, the intensity of 40 µT at a 76.6 Hz signal combined with the continuous magnetic field is identified with the magnetic field sensor 35.3 mm ± 1.09 mm far from the coil, while the intensity of 20 µT at 0 Hz is identified with the sensor distance of 31.2 mm ± 1.15 mm.

These values are important to verify the applicability of the technique in a real scenario, considering the average dimensions of the patient’s wrist. As the human wrist has an average width of 65.0 mm, with its largest area filled with bone tissue, the most central part of the bone marrow is, on average, 32.5 mm away from the human skin [55]. The bone surface, however, has a short distance from the epidermis, the most superficial and external region of the skin, given the low concentration of fat and muscle tissue in the wrist joint region. Furthermore, the inner fixation mesh, on which the stimulation coil is positioned, has 2.0 mm of thickness. The coil is placed over the outer immobilization mesh with its supply wires connecting it to the stimulation device, which can be tied to the arm or waist.

It is noteworthy that the application of a magnetic field consistent with the characteristics of CMF for a low power system and small coil dimensions (2.5 cm in diameter) was not found in a similar system integrated in an orthosis, as presented in this study. Equipment purchased commercially for orthopedics that use CMF mapped in the background have dimensions higher than 28 cm and their application is not concomitant with immobilization. Previous works have been dedicated to seeking an orthosis design that uses smart materials suitable for self-adjustment to the patient’s anatomical and functional changes, and supports dynamic activities (e.g., physical therapy for people with wrist and hand atrophy). However, these do not have active systems to assist in the rehabilitation process through the use of stimulation [49,56,57,58].

Few studies have effectively integrated stimulation systems with orthoses and, for those that integrate such resources, aspects of difficult technological incorporation for clinical use were mapped [59,60]. It is important to highlight that none of the studies searched so far proposed the integration of a CMF electrostimulator device to immobilization material in orthopedic therapy. Mohammadi, M. et al., 2022 define, in a review of 4D-printed soft orthoses, that despite recent advances in the area, orthoses with electroactive systems have low adherence by patients and physicians, mainly due to the inadequacy of the treatment routine because of inappropriate weight and shape. The authors emphasize that the main challenge is to obtain a solution that is compact and has features for practical use, especially for hand fractures [59,60].

The use of an CFM electrostimulator is advantageous, because among the known techniques of magnetic stimulation, this is the one that has the greatest effectiveness with the shortest time of use [16,54]. However, in the literature, CFM generators are laboratory devices, just as commercial magnetic stimulation devices are large. The difficulty in using the electrostimulator concomitantly with the orthosis and in adapting to the patient’s daily use are considered major obstacles to its widespread use [29]. Hence, the orthosis model with two meshes and small electromagnetic transducer proposed in this work provides a solution for these scenarios.

Regarding the mobile application and remote database, their functions have potential as a way to engage the patient in the treatment and provide health professionals with mechanisms for a more accurate follow-up of the treatment. In the clinical aspect, the system allows the healthcare professional to monitor the number of applications made and adjust the frequency of operation. In this case, the possibility of adjusting the frequency may have potential for future research where different electrostimulation parameters might be compared in order to define the one that best suits the treatment. In terms of usability, the application alerts users of the correct time to apply electrostimulation and confirms the proper completion of electrostimulation after 30 min. Precisely because all values will be recorded in the application/database, the solution can provide real-time information to the health professional and ensure patient continuity in treatment. The recording of data in the internal memory of the microcontroller and in the remote database adds to the technology’s greater robustness in the treatment of data, avoiding the loss of the patient’s medical history even if the circuit has operation failures. Moreover, this feature is important so that, in future research projects, computational aspects, including artificial intelligence and machine learning, could be considered for precision and differentiation in the treatment of each patient [58].

Thus, considering the above, the solution investigated moves the advances obtained by scientific work closer to their practical application in medicine. The orthosis has characteristics that are superior to those of traditional techniques, such as orthopedic plaster and fiberglass, because it prevents excessive sweating by reducing dermal bacterial proliferation, facilitates the definition and adjustment of fracture fixation points during treatment and provides an active resource in reducing bone regeneration time. Likewise, it points to innovative aspects in its functionality, using materials and manufacturing technologies that are easily accessible and widely investigated in the literature. Thus, the work takes advantage of advances in additive manufacturing, especially of PLA and TPU by FDM printing, and in the use of tissue stimulation, with emphasis on CMF, to propose a step forward. This step focused on including features that facilitate the incorporation of the solution into medical practice, such as those highlighted in this paper.

## 5. Conclusions

From the results achieved from the outlined methodology, it is possible to conclude that the developed system has advantageous characteristics compared to those identified in orthopedic immobilization and bone regeneration technologies currently used. The orthosis design divided into two polymeric meshes with different characteristics has the potential to solve immobilization and stabilization restrictions of bone fracture fixation points, mapped in the scientific literature as disadvantages to the use of the additive manufacturing technique for this application.

The generation of the magnetic field controlled by mobile application presented results compatible with those identified as safe and effective for therapeutic applications of electrostimulation in the scientific literature, and the characteristics, properties and formats of the CMF electrostimulator and software indicate advantageous aspects not identified in previous research projects. Thus, the results use widely accessible materials and techniques, such as PLA and TPU polymers as well as an FDM printer, to advance scientific concepts so that they could be more easily incorporated into the clinical environment.

Therefore, the following innovations proposed by this study stand out: the orthosis is superior in terms of ease of use and risk reduction when compared to orthopedic plaster and fiberglass (e.g., dermatitis, ease of following-up the treatment progress, mechanical stability of the fracture area, avoidance of deep vein thrombosis); addresses a solution in the technique to avoid mobility of the fractured wrist joint; allows simultaneous immobilization and stimulation using a small device by CMF, an aspect not identified in previous works; it was tested for exposure to chemical agents present in the patient’s daily life, having achieved an appropriate response; and it is controlled by a mobile application with a remote database, which allows real-time monitoring of the doctor and opens the possibility for new computational resources to be used in the future.

## Figures and Tables

**Figure 1 materials-15-08141-f001:**
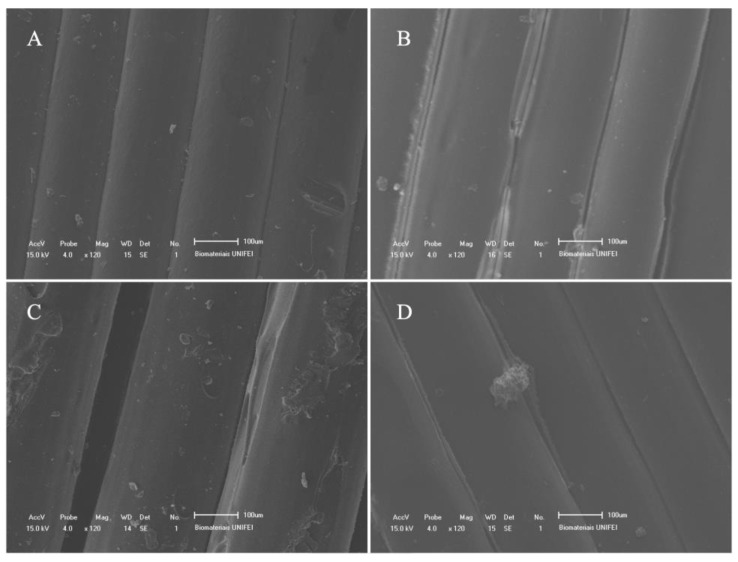
SEM of polymer samples for (**A**) ABS exposed to water; (**B**) TPU exposed to sweat; (**C**) TPU exposed to alcohol and (**D**) PLA exposed to soap.

**Figure 2 materials-15-08141-f002:**
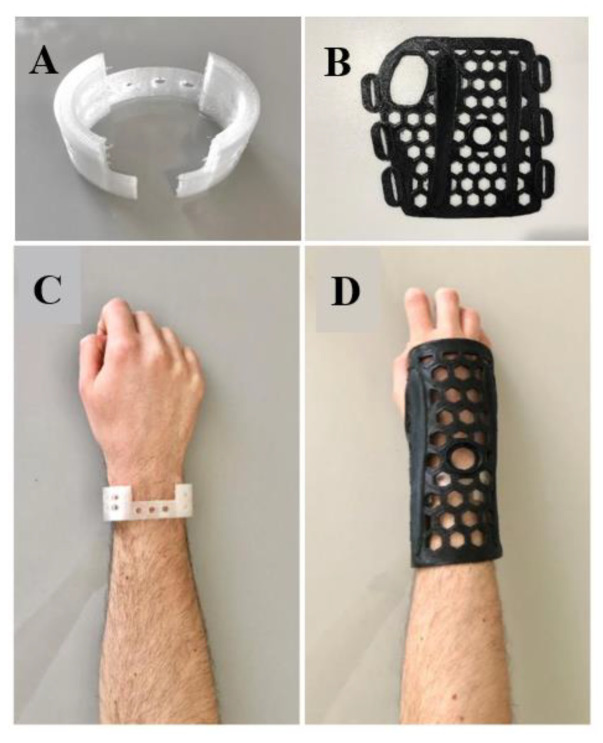
Modeling of the meshes printed on the wrist of a person to illustrate the way of use, where (**A**) it is the inner fixation mesh printed; (**B**) the outer immobilization mesh printed; (**C**) only the placement of the inner fixation mesh and (**D**) the joint use of the fixation mesh with the immobilization mesh.

**Figure 3 materials-15-08141-f003:**
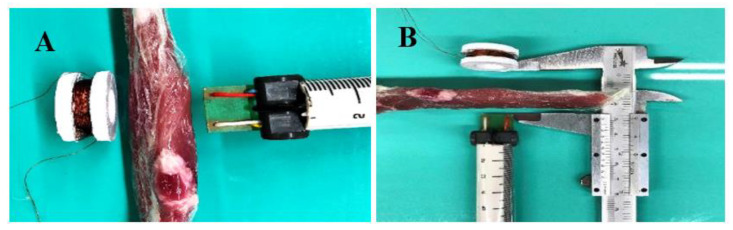
Evaluation of magnetic permeability of the magnetic field by biological tissue, with (**A**) coil on the left, biological tissue in the center and tip of the field analyzer on the right; In (**B**), the caliper used to measure the distance from the coil to the analyzer.

**Figure 4 materials-15-08141-f004:**
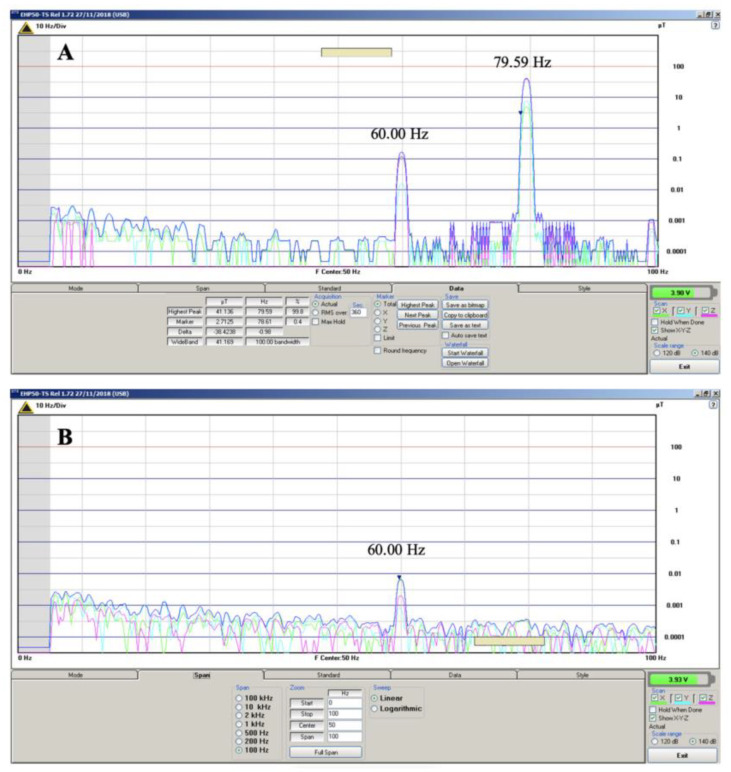
Intensity of the magnetic field generated by the small electrostimulator, (**A**) presence of the fields generated by the device and the scattered electric grid and (**B**) the field effect of the scattered electric grid.

**Figure 5 materials-15-08141-f005:**
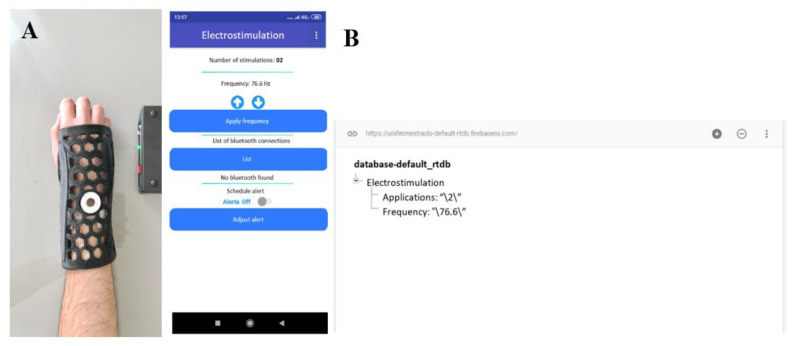
Representation of the system as a whole, (**A**) orthosis integrated with the stimulator controlled by the app; (**B**) mobile app interface with database records highlighting the number of electrostimulation applications and magnetic field frequency.

**Table 1 materials-15-08141-t001:** Records of magnetic field values for stimulation in the coil by alternating and continuous electrical signal with readings of signal capture distances.

Scenario	Intensity of Magnetic Field (µT)	Distance (mm)
Alternating magnetic field (76.6 Hz), combined with continuous magnetic one, peak voltage (V_p_) of 12V	40	35.3 ± 1.09
	20	39.2 ± 1.15

## Data Availability

Not applicable.

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
