# Peer review of "Polymeric Orthosis with Electromagnetic Stimulator Controlled by Mobile Application for Bone Fracture Healing: Evaluation of Design Concepts for Medical Use"

_materials, 2022, doi:10.3390/ma15228141_

Round 1
Reviewer 1 Report
Th research "Polymeric orthosis with electromagnetic stimulator controlled by mobile application for bone fracture healing" has been poorly presented. The study design and methods are not clear at all, and the results have not been presented properly.
In methods, What is the basis of the design? What are the dimensions or key components? What is new in the material used? PLA/ABS/TPU 3D printing has no novelty, in terms of material. Bovine tissue is not the same as human tissue-should have had a reference on why this was chosen and the test protocol. How many patients were part of the study? How was data analysis conducted?
In results, why some final numbers or percentages were only declared, where are the test result plots? Some test setup was shown, but no results were reported. The mobile application screen displayed in Figure 4 has limited purpose, and shows just inputs, no outputs.
In discussion, no comparisons with existing technologies have been included.
The conclusion states improved results, but no results or comparisons were reported throughout the manuscript.
Author Response
Dear Reviewer,
I and the other authors of the manuscript are very grateful for your comments. Please, find enclosed in the PDF file our considerations on the adjustments made in the paper.
If you have any questions, please do not hesitate to contact me.
Sincerely,
Patricia Capellato.

Reviewer 2 Report
Please, see the attached file.

Author Response

(The authors gave the same response as above.)

Reviewer 3 Report
Dear Authors, I have reviewed the manuscript "Polymeric orthosis with electromagnetic stimulator controlled by mobile application for bone fracture healing". The manuscript describes the design and fabrication of a novel 3D printed orthotic device, equipped with US capabilities. The study design and methods are well chosen and carefully explained. In terms of material science, the findings are not novel, however, they highlight important practical aspects. The overall topic is undoubtedly interesting, but in my opinion, some questions must be adressed:
1.) In the introduction, the overall presentation of upper limb bone fractures are not complete - there is no information regarding of incidence or prevalence. I recommend the sources of the following recent article:
Schlégl, Á.T.; Told, R.; Kardos, K.; SzÅ‘ke, A.; Ujfalusi, Z.; Maróti, P. Evaluation and Comparison of Traditional Plaster and Fiberglass Casts with 3D-Printed PLA and PLA–CaCO3 Composite Splints for Bone-Fracture Management. Polymers 2022, 14, 3571. https://doi.org/10.3390/polym14173571
2,) Why did the research team have chosen ABS insted of PLA, which is a clearly biocompatible material? ABS is not considered as a biocompatible material.
3.) The authors should present the advantages and disadvanteges of 3D printing compared to traditional casting/orthoses in more details, in the discussion/conclusion section.
4.) In the Methods, the design process of the orthosis is not fully described. How did the researchers created the 3D model for 3D printing, in details?
5.) I miss supporting materials (e.g. 3D printing files, questionnare etc.) from the Supplementary / Data Repository. This would be important in terms of reproduction of the research work.
6.) The novelty of findings should be more emphasied in the conclusion.
Author Response

(The authors gave the same response as above.)

Round 2
Reviewer 1 Report
All my comments have been responded to and the reviewer feels that significant changes were made by the authors to improve the manuscript.
Author Response
Dear reviewer,
We greatly appreciate your contribution to the improvement of this manuscript. Undoubtedly, your comments helped us to make the text clearer.
Thank you for your availability to review.
Reviewer 2 Report
I would like to thank the authors for responding point by point to the comments and for their work in revising the paper, which further improved its overall quality.
I would just like to make a couple of comments on the revised part:
1) Regarding the sentence in the abstract "Results: degradation was observed after exposition of the chemical agents." I think it contains an error. It should be "Results: no degradation was observed after exposition of the chemical agents."
2) Regarding the questionnaires (Annex A), it would be appropriate to also indicate the range of the expected score (was a 5-point scale used? This is to aid interpretation of mean and standard deviation results). Also, some questions report the mean score but not the standard deviation: for uniformity with the other questions, it would be appropriate to add them.
Author Response
Dear reviewer,
We greatly appreciate your contribution to the improvement of this manuscript. Undoubtedly, your notes were fundamental to make the text clearer.
About the two points you highlighted in the last review, they have been fixed. In the abstract, the sentence on material degradation has been changed. In annex A, information on the question scores and missing standard deviation markings was included.
Thank you for your availability for the review.
Reviewer 3 Report
Dear Authors, thank you very much for answering all of my questions. With the corrections, my opinion is that the article can be published in Materials.
Thank you very much. I wish you all the best.
Author Response
Dear reviewer,
We greatly appreciate your contribution to the improvement of this manuscript. Undoubtedly, your comments helped us to make the text clearer.
Thank you for your availability to review.
We also wish you all the best.